# An experimental assessment of detection dog ability to locate great crested newts (*Triturus cristatus*) at distance and through soil

**Nicola Jayne Glover**[1,2]*, **Louise Elizabeth Wilson**[3], **Amy Leedale**[1], **Robert Jehle**[1]

**1** School of Science, Engineering and Environment, University of Salford, Salford, United Kingdom,
**2** Wessex Water, Bath, United Kingdom, **3** Conservation K9 Consultancy, Wrexham, United Kingdom

* n.glover@edu.salford.ac.uk

**Data Availability Statement:** Please use the following link to our raw data on Dryad: https://doi.org/10.5061/dryad.sf7m0cg9t.

## Abstract

Detection dogs are increasingly used to locate cryptic wildlife species, but their use for amphibians is still rather underexplored. In the present paper we focus on the great crested newt (*Triturus cristatus*), a European species which is experiencing high conservation concerns across its range, and assess the ability of a trained detection dog to locate individuals during their terrestrial phase. More specifically, we used a series of experiments to document whether a range of distances between target newts and the detection dog (odour channelled through pipes 68 mm in diameter) affects the localisation, and to assess the ability and efficiency of target newt detection in simulated subterranean refugia through 200 mm of two common soil types (clay and sandy soil, both with and without air vents to mimic mammal burrows, a common refuge used by *T. cristatus*). The detection dog accurately located all individual *T. cristatus* across the entire range of tested distances (0.25 m– 2.0 m). The substrate trials revealed that the detection dog could locate individuals also through soil. Contrary to existing studies with detection dogs in human forensic contexts, however, detection was generally slower for *T. cristatus* under sandy soil compared to clay soil, particularly when a vent was absent. Our study provides a general baseline for the use of detection dogs in locating *T. cristatus* and similar amphibian species during their terrestrial phase.

## Introduction

In order to complete their development, many temperate amphibian species combine waterbodies for breeding and larval development with a terrestrial phase for aestivation and hibernation [1]. The aquatic phase is generally rather well documented, as individuals can be recorded with relative ease at high concentration for example in ponds [2, 3]. When individuals disperse over wider areas onto land, however, they often lead a secretive lifestyle and are more difficult to find [4, 5].

The great crested newt (*Triturus cristatus*) is a urodele amphibian species which occupies parts of central and northern Europe including the United Kingdom, and in most parts of its range is rapidly declining primarily due to the widespread loss of suitable habitat [6]. As a

**Funding:** The author(s) received no specific funding for this work.

**Competing interests:** The authors have declared that no competing interests exist.

result, *T. cristatus* is afforded the highest protection under European (Regulation 41 of the Conservation of Habitats and Species Regulations 2010; Annex 4 of the Fauna, Flora and Habitat (FFH) Directive) and UK (Section 9 of the Wildlife and Countryside Act 1981) legislation. It therefore currently experiences a disproportionally high level of attention from conservation practitioners, including for example large-scale management schemes based on data derived from environmental DNA and initiatives for habitat creation in peripheral parts of its range [7, 8]. However, as typical for other amphibians with bi-phasic life cycles, rather little is known about its terrestrial habitat use. While systematic terrestrial hand searches are part of standard protocols for practical conservation measures [9], their efficiency strongly depends on the local habitat and is generally rather low, likely because of the common use of subterranean refuges which are difficult to reach (e.g. [10]). Approaches such as based on radio- and fluorescent pigment-tracking have revealed important information on local migration distances and refuge use after breeding [10–12], but are time-consuming to conduct, restricted to a limited number of individuals whose welfare can be compromised, and can be applied over only limited durations such as days or weeks.

Due to their olfactory capabilities, wildlife detection dogs are increasingly deployed around the world to assist in locating cryptic animal species in their natural environments [13–16]. Depending on the breed, a dog's sense of smell is estimated to be 1,000 to 10,000 times better than that of humans, corresponding to a relative area of the brain for olfactorial processing which is about forty times larger [17]. As a consequence, the performance of trained detection dogs in wildlife searches is generally 4–12 times better than that of experienced human surveyors, which in contrast to dogs predominately rely on visual and/or auditory cues [18–24]. The olfactorial capabilities of detection dogs also allow them to find elusive species in a variety of environments where technological devices to locate them are either unavailable or less efficient [24, 25].

Wildlife detection dogs would offer an opportunity to locate individual amphibians during the terrestrial phase of their life, but their use currently remains rather underexplored compared to other taxonomic groups such as reptiles and mammals [26–31]. At present, projects involving detection dogs have only been conducted on a small number of amphibian species globally (e.g. North American salamanders: *Amystoma californiense*, *Plethodon neomexicanus*; South African giant bull frogs *Pyxicephalus adspersus*, Australian baw frog *Philoria frosti;* see [16, 32–34]). In Europe, a recent study found that detection dog teams were effective at locating both *T. cristatus* and smooth newts (*Lissotriton vulgaris*) in a variety of above ground surface refuges in tall and short grassland, as well as open and dense woodland habitat [35]. The study found that, in an open environment, *T. cristatus* has a scent profile which is detectable over a range of approximately 0.2 m, although factors such as habitat type, weather, and type of search had an influence on detection probability [35].

Given the current distinct need to increase our understanding on how environmental factors may influence the probability of detection, we here conducted a set of experiments to document whether a detection dog can olfactorily locate individual *T. cristatus* (i) at a range of distances where odour is allowed to move along a channelled direction through pipes, and (ii) through two different soil types in simulated subterranean refugia with or without vents to mimic mammal burrows.

## Materials and methods

### Detection dog/handler team

All experiments were undertaken with an English springer spaniel ("Freya") born on the 25[th] of August 2015 and handled by the first author (NG). The dog/handler team has been trained

using positive reinforcement since 2017 by Conservation K9 Consultancy (founded by the co-author LEW), receiving annual assessments involving shadowing by LEW during operational searches and undertaking controlled field assessments to determine whether the team is performing to Conservation K9 Consultancy standards. Training to locate individual *T. cristatus* started in 2018, and a discrimination assessment (conducted in 2020 and as outlined in [36]) confirmed that Freya is able to distinguish between *T. cristatus* and all other common amphibians native to the UK (*Bufo bufo*, *Rana temporaria*, *Lissotriton helveticus*, *L. vulgaris*) obtaining a score above the required 80% threshold for successful discriminations. Positive indications are signalled by Freya through lying or sitting down. All experiments were approved through licences issued by Natural England, a UK governing body (licence numbers 2019-39743-SCI-SCI; 2020-45697-SCI-SCI; 2021-51164-SCI-SCI).

## Channelled distance perception trials

Sixteen *T. cristatus* (11 males and five females) were captured from a quarry and used for a range of trials carried out in Westerleigh, South Gloucestershire, England (Ordinance Survey (OS) grid reference: ST695795) between July and September 2020. All individuals were weighed to the nearest 0.1 g. Weights ranged from 7.0 g to 11.0 g. A 0.5 m x 12.5 m rig made of plastic-coated wood was constructed with eight holes spaced 1 m apart to hold plastic guttering pipes with a diameter of 68 mm (Fig 1A). At the end of the pipes, they were connected to a plastic container holding individual *T. cristatus* fitted with a lid containing fifteen 3 mm air holes at consistent patterns to allow the scent to escape. Planks of wood ensured an upright position of the rig, and a stake was placed behind the containers to prevent them from slipping during the trial (Fig 1B). Pipes ranging from 0.25 m to 2 m in length were used. Two runs across the eight-pipe rig were conducted for each pipe length with a single target newt in one of the eight pipes, successively increasing the length by 0.25 m and resulting in a total of 16 trials. The position of the board was identical for each run. Due to licence restrictions limiting the number of individual newts available for this assessment, we were unable to conduct additional repetitions.

The experiment took place on the 20th of August 2020 and involved three testers, the dog/handler team, and an observer, and were undertaken double blind, i.e. the dog/handler team

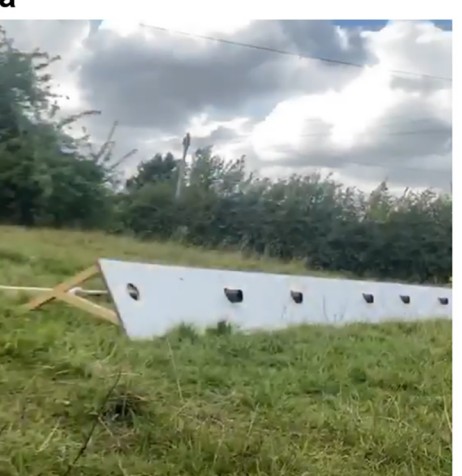
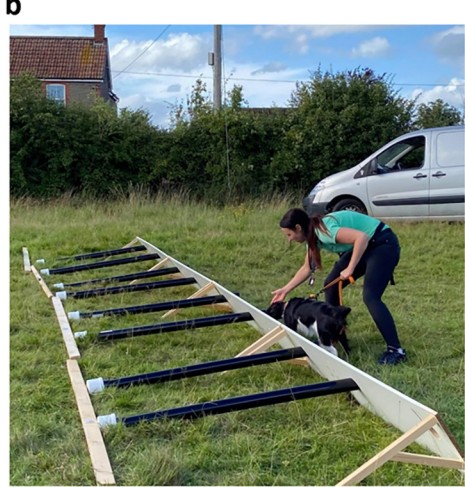

**Fig 1. Channelled distance perception rig setup.** (a) Channelled distance perception rig setup viewed from the front. For details see text. (b) View of the rig from the back with detection dog Freya and handler Nikki Glover moving along front of the rig checking each pipe entrance.

and observer were out of sight when the testers set up each run and the testers were out of sight during the runs. This type of setup is carried out to avoid unconscious signalling from the handler and observer/tester to the detection dog [37]. A random number created in Microsoft Excel was used to determine at which position along the rig the target newt was placed. Following the Centre for the Protection of National Infrastructure (CPNI) Canine Odour Discrimination Test guidelines [38], Position 1 and 8 of the rig were not used as these locations were likely to yield misses or false indications relating to procedural training issues [39]. Position 1 acts as an overlap and ensures that the detection dog has settled into the search prior to encountering the first potential target, and Position 8 addresses indications given by dogs that have failed to find a target and take the chance of getting rewarded at the last location. Before trials, *T. cristatus* were sprayed with untreated water to keep the skin moist and placed in separate containers at least 30 minutes prior to the start of the search. The dog/handler team positioned themselves 2–3 m in front of the first pipe before performing a systematic check of all pipes on a lead with the handler asking the dog to check each pipe by signalling with their hand (Fig 1B). The handler then called out the container number when the dog indicated, confirmed with the testers who were out of view. One additional repeat (starting from pipe Location 1) was allowed if the detection dog did not indicate on the first run. If the dog indicated correctly on *T. cristatus*, then the dog was rewarded and the trial was ended. After the first eight trials covering four pipe lengths (0.25 m– 1 m), a 30-minute break was allowed before continuing with a further set of trials using pipes at length ranging between 1.25 m and 2 m. General weather conditions were noted during each trial run as sunny (0% clouds visible), sunny intervals (cloudy with intermittent sunshine), cloudy (100% cloud coverage with no sun visible), as well as dry (no precipitation), drizzle (light precipitation), rain (moderate precipitation), and heavy rain (torrential precipitation). In addition, air temperature and humidity were measured using a hygrometer thermometer, wind speed was measured via an anemometer using the Beaufort scale and wind direction assessed by scattering baby powder; cloud cover was estimated as percentage of clouds in the sky.

In addition to correct or false indications, general detection dog behaviour was recorded during the trials and included motivation, stamina, focus, cooperation, fatigue, distractibility, and arousal. Handler behaviour was also recorded and included ability to read the dog's behaviour, cooperation, motivation, focus, stamina and overall interaction with the dog. Fresh vinyl gloves were used to set up each trial to avoid scent contamination. The tester handling the individual *T. cristatus* was restricted to undertaking this task alone and refrained from touching the outer edge of the container to avoid contamination. The dog/handler team trained for two weeks in advance of the assessment, receiving consultation from Conservation K9 Consultancy.

## Soil interference trials

Thirty *T. cristatus* (16 males and 14 females) were retrieved from a quarry in south Gloucestershire between July and September 2021. All *T. cristatus* used for the experiments were individually weighed to the nearest 0.1 g and measured from the tip of the snout to the end of the cloaca to the nearest mm (see S1 Fig). Two soil types were used: sandy soil collected from Calne Without, Bowden Hill, Wiltshire, Southwest England (OS grid reference ST954684), and clay soil collected from Broad Lane, Westerleigh Road, Westerleigh, South Gloucestershire, Southwest England (OS grid reference: ST695795). For soil characterisation, three reference samples were obtained from each site using a 200 mm long core and merged into one sealed plastic bag for analysis at the University of Salford. In order to measure the pH, 20 ml of fresh soil was placed into a clean 50 ml beaker. Distilled water was added to the 40 ml mark,

and the mixture was left on an automatic stirrer for 15 minutes. The probe of a calibrated pH meter was then placed into the sample and left for 30 seconds for the reading to settle. This was repeated three times for each soil type, with the probe cleaned with distilled water between each reading. To gauge moisture content for each soil type, the soil was weighed, oven dried overnight at 105˚C, and re-weighed. The moisture content was obtained by subtracting the dry weight from the wet weight, divided by the dry weight and multiplied by 100. Soil texture was analysed using an LA-960 Slurry Sampler, which uses laser diffraction to measure light scattered from the soil particles as it passes through the measurement cell. The percentage of silt, clay and sand was determined using a soil texture chart.

The soil interference trials were carried out in the same location as the channelled distance perception trials. Four plots measuring 10 m x 4 m were erected 2m apart and demarcated using barrier tape. Each plot had eight holes dug to 250 mm depth and spaced evenly 2 m apart in a 2 x 4 setup (Fig 2A). This was so that all pipes were flush to the ground (Fig 2B). Compartments to house individual *T. cristatus* were made from a guttering pipe socket (50 mm high, 61 mm in diameter), which was considered to supply enough air to the target newts during the trial. A plastic bung was placed at the base of the compartment to keep individuals in place and to reduce the chance of residual scent being left in the hole following the trial. A fibreglass 4 mm by 4 mm mesh was placed at the top of the compartment, and a 200 mm plastic guttering pipe with 68 mm diameter was slotted over the top of the pipe socket, holding the netting in place (Fig 3A). The pipe was weighed prior to the newt entering the compartment, and assigned to one of four treatments: Sandy Soil Full, Sandy Soil Vented, Clay Soil Full and Clay Soil Vented (Fig 3B). The vented treatments comprised a plastic-coated wire mesh with 4 x 4 mm gaps. The mesh was cut to 200 mm length to sit within the 200 mm long pipe. The mesh was rolled up into a cylindrical shape and secured with cable ties with a diameter of approximately 25 mm to represent small mammal burrows (a common subterranean refuge of *T. cristatus* ([11], see also [40] for the closely related *Triturus carnifex*). The soil was placed into the pipes using a trowel with soil placed around the wire mesh vents to ensure they stayed open (Fig 3B). The pipes containing soil and newts were weighed to provide the weight of the soil by subtracting the pipe weight and the weight of the individual *T. cristatus* used.

Locations of each pipe within the plots were allocated using a random number generated in Microsoft Excel. Pipes with all treatment types and blank compartments served as controls. Two pipes of every treatment type were present in each plot with one soil type (e.g. clay)

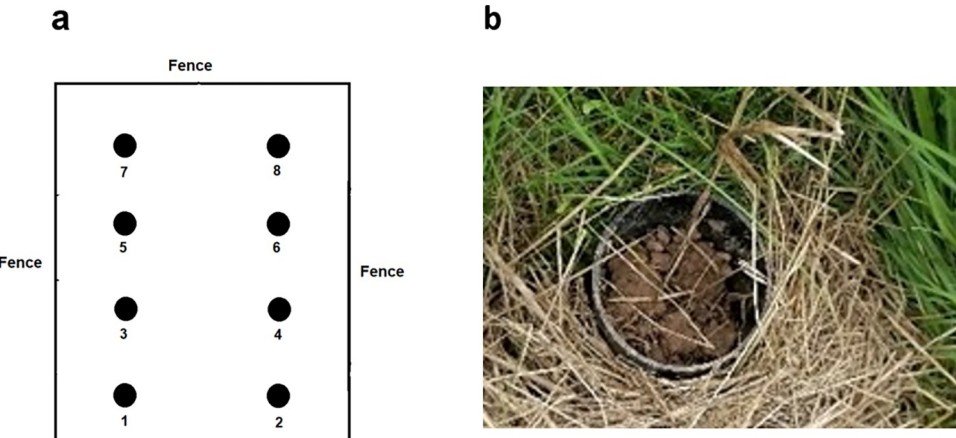

**Fig 2. Soil interference trial plot setup.** (a) Location of pipes within a single plot (b) simulated terrestrial subterranean refuge flush with ground.

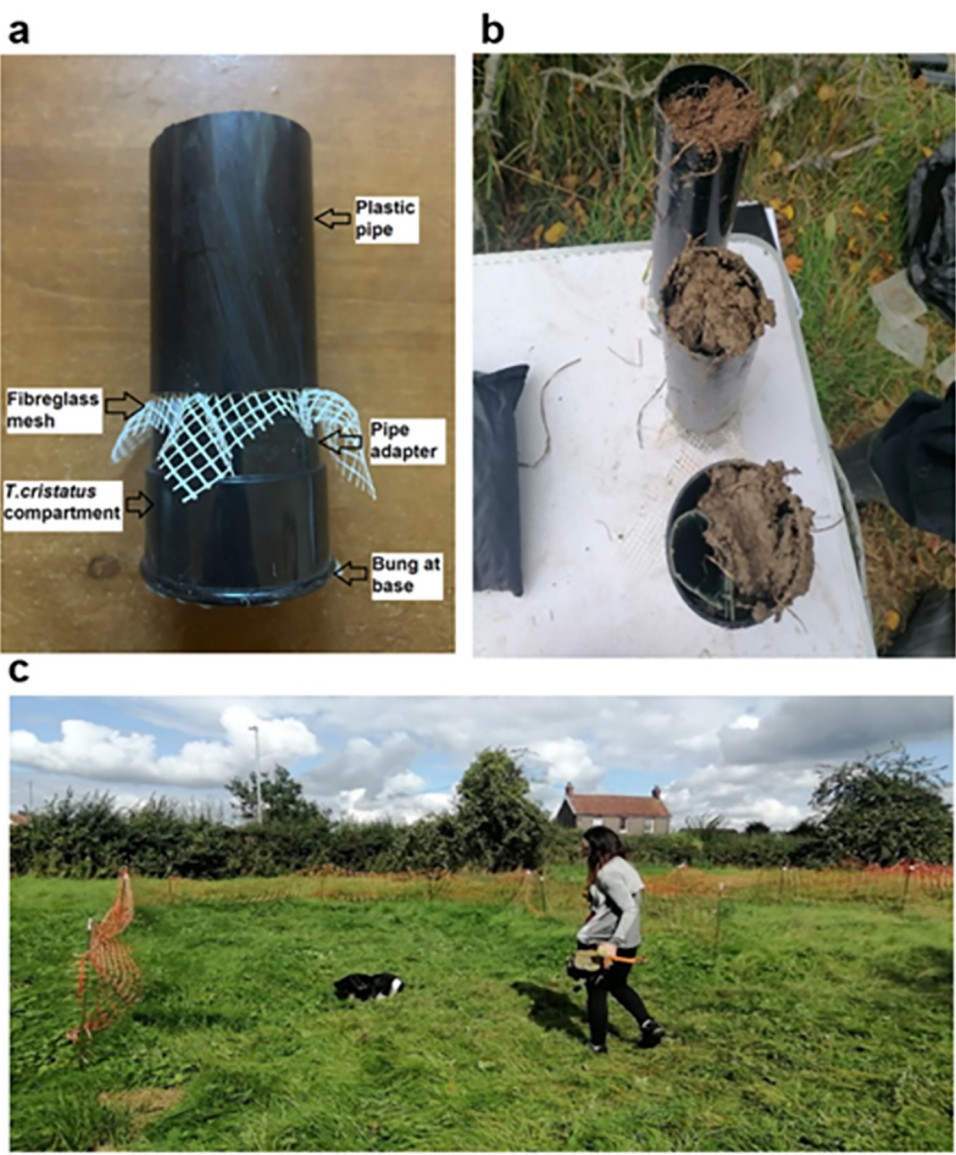

**Fig 3. Simulated terrestrial subterraneous refuge setup.** (a) diagram displaying pipe setup. For details see text (b) Sandy Soil Full treatment at back, Clay Soil Full in middle and Clay Soil Vent at the front. (c) The detection dog/handler team at work in a designated plot. Detection dog Freya indicating on correct tube with *T.cristatus* present 200mm below Clay Soil Full treatment type. Handler about to reward detection dog with tennis ball.

located to the left of the plot and the other soil type (e.g. sandy) positioned to the right. The location of each soil type changed between each run. Vented and full treatments were positioned alternatively within each plot. Three out of the four plots contained a single *T. cristatus* with one plot serving as a control. All pipes were placed out 30 minutes prior to the start of the trial to allow the scent to penetrate through the pipe. This was considered enough time to allow the scent to flow through 200 mm of soil whilst also maintaining the welfare of the target newts.

The experiments involved one or two assessors and the dog/handler team. Unlike the channelled distance perception trials, the experiments were undertaken blind as opposed to double blind due to limited numbers of assessors available. The assessor therefore performed the role

of the observer and the tester and was present during the test runs. The dog/handler team was not aware of the location of the *T. cristatus* in each plot. During each trial, the assessor measured weather conditions using same methodology as described for the channelled distance perception trials. Soil temperature and moisture content was measured by placing the probe 50 mm into the soil using a Soil Condition Meter DSMM600, with moisture levels displayed as <5% (Dry+), 5 to 10% (Dry), 10 to 20% (Normal), 20 to 30% (Wet) and >30% (Wet+). Soil moisture measurements were taken once before each 4-plot trial took place to minimise disturbance. It took a maximum of 10 minutes to conduct searches across Plots 1–4, rendering it unlikely for moisture and temperature to change drastically.

The handler positioned the dog at the entrance of the plot, unclipped the lead and asked the detection dog to perform a search of a single plot. The handler maintained position at the entrance of the plot and monitored the detection dog's behaviour during the search. When the dog indicated (Fig 3C), the handler confirmed the location with the assessor and if correct the dog was rewarded, followed by a two-minute play session. If the dog/handler team was incorrect, then the search in each plot continued, noting down the number and locations of false indications until the correct indication was given. The trial was terminated if the dog/handler team could not locate the correct pipe within 180 seconds and the handler could not determine if the plot was blank. This length of time was considered sufficient given the size of the plot and the time taken to locate correct pipes during the practice sessions. The handler called 'blank' before the 180 seconds if she believed the plot did not have a *T. cristatus* individual present due to no change in behaviour exhibited by the detection dog. Once the dog/handler team had completed a plot search they would move onto the next plot and repeat the above method. A 45-minute break was given between each 4-plot trial. New pipes were used during each trial. Fresh gloves were worn by the assessor when placing newts into the compartments to avoid transferring newt scent onto the outside of pipes. All pipes were moved to new positions between each trial. Each 4-plot trial was repeated four times within one day, and trials were carried out over two consecutive days in four successive weeks (19th August 2021 and 17th September 2021), resulting in a total of 128 single-plot trial runs. Twelve individual *T. cristatus* were selected at random to be used for the trial day, placed in a labelled container to prevent re-use. Twelve additional individuals were used for the consecutive day, taken from the tanks at random. The dog/handler team trained for three weeks in advance of the soil trial assessment, receiving consultation from Conservation K9 Consultancy. As measures of performance by the dog/handler team, numbers and locations of false negative, false positive, true negative and true positive indications were noted.

## Statistical analyses

Due to limited repetitions undertaken for the channelled distance perception trials and a universally high success rate (see below), we refrained from detailed statistical analyses of the data obtained in this experiment.

For the soil interference trials, a chi-square test was conducted to determine whether false indications occurred more frequently in blank plots compared to plots which contained a *T. cristatus* individual. Within plots containing a *T. cristatus* individual, the effect of treatment on the frequency of false indications was analysed using a binomial General Linear Mixed Model (GLMM), with correct/false indication (0/1) specified as a response variable. Successful trials were further analysed using a restricted maximum likelihood (REML) GLMM with log-transformed time to detection as the response variable. Initial GLMMs included treatment and sex as fixed effects, with weight, air temperature and humidity as random effects. The most suitable GLMMs were subsequently selected based on the Akaike information criterion (AIC) and

backwards-stepwise model refinement, in which non-significant terms were sequentially removed. For false indications, the final model included treatment as a fixed effect, with no random effect terms specified. For detection time, the final model included treatment as a fixed effect and air temperature as a random effect. Where a significant effect of treatment was identified, a Tukey HSD test was carried out on the estimated marginal means from the GLMM, to determine pairwise contrasts between all treatments *posthoc*. All tests were conducted in the software *R* [41].

## Results

### Channelled distance perception trials

Trials were carried out between 14:20h and 16:30h. Air temperature ranged between 27.8˚C at the start of the trial and 21.6˚C at the end of the trial, and air humidity ranged between 43% at the beginning of the trial and 60% by the end of the trial. Wind speed ranged between 3 Beaufort for pipe lengths 0.75 m—1.5 m and 5 Beaufort for pipe length 1.75 m—2 m. The wind remained easterly throughout the trial, and cloud cover varied between 15% and 30% (see S1 Table).

The dog/handler team was able to successfully locate individual *T. cristatus* across all measured distances, at a total of 87.5% of runs. Two false positive indications at initial runs occurred at lengths of 0.25 m and 0.75 m, with correct indications during second runs. No non-indications took place (see Table 1 for the outcome of all 16 trial runs). False indications were only recorded towards the beginning of the trial. This assessment coincided with the observation that the detection dog/handler team maintained motivation throughout the trial, with stamina and concentration increasing with the ongoing assessment.

### Substrate interference trials

The pH of the sandy and clay soil reference samples was 6.38 and 6.68, respectively. Wet and dry weights for sandy soil were 20.03 g and 18.20 g, respectively, at a moisture content of 10%,

**Table 1. Results of the channelled distance perception trial.** X indicates the randomised location of a great crested newt (*Triturus cristatus*). c: correct, fp: false positive.

| Distance cm) | Pot 1 | Pot 2 | Pot 3 | Pot 4 | Pot 5 | Pot 6 | Pot 7 | Pot 8 | Attempt 1 | Attempt 2 |
|---|---|---|---|---|---|---|---|---|---|---|
| 25 | | | | | | X | | | 5 (fp) | 6 (Correct) |
| 25 | | | | | | | X | | 7 (c) | N/A |
| 50 | | X | | | | | | | 2 (c) | N/A |
| 50 | | | | | | X | | | 6 (c) | N/A |
| 75 | | | | | | | X | | 6 (fp) | 7 (correct) |
| 75 | | | | X | | | | | 4 (c) | N/A |
| 100 | | | X | | | | | | 3 (c) | N/A |
| 100 | | | | | X | | | | 5 (c) | N/A |
| 30-minute break | | | | | | | | | | |
| 125 | | | X | | | | | | 3 (c) | N/A |
| 125 | | | | | X | | | | 5 (c) | N/A |
| 150 | | | | | | X | | | 6 (c) | N/A |
| 150 | | | | | | | X | | 7 (c) | N/A |
| 175 | | | | X | | | | | 4 (c) | N/A |
| 175 | | X | | | | | | | 2 (c) | N/A |
| 200 | | | | | | X | | | 6 (c) | N/A |
| 200 | | | X | | | | | | 3 (c) | N/A |

**Table 2. Outputs from a binomial GLMM of the effect of soil treatment on frequency of false indications.**

| Treatment | Estimate ± SD | z | p |
| --- | --- | --- | --- |
| Clay Soil Full; Intercept | 3.09 ± 1.02 | 3.02 | <0.01 |
| Clay Soil Vent | 0.09 ± 1.44 | 0.06 | 0.95 |
| Sandy Soil Full | -1.48 ± 1.56 | -1.28 | 0.20 |
| Sandy Soil Vent | -0.69 ± 1.26 | -0.55 | 0.58 |

with the corresponding values for clay soil being 20.37 g and 15.96 g with a moisture content of 27%. Sandy soil contained 55% sand, 44% silt and 0.81% clay (classified as a loamy sand). Clay soil contained 25% sand, 45% silt and 30% clay (classified as a clay loam).

Air temperature during trials ranged between 14.3 and 26.1˚C with an average of 19.6˚C. All trials were completed by 1 p.m. Soil temperatures were on average 1˚C lower than air temperatures, without temperature differences between clay and sandy soil. Air humidity ranged between 44% and 98% with an average of 70%. Air temperatures were negatively correlated with humidity (linear regression, r = 0.51). Clay soil had moisture content varying between Wet (20–30%) and Wet+ (>30%), and sandy soil had a moisture content of Dry (5–10%) or Dry+ (<5%). Wind speed varied between 0 and 3 Beaufort, at varying wind directions from southwest, west, northwest, northeast or south which however were consistent between single-plot runs in each 4-plot trial. Cloud cover varied between 30% and 100%. Weather conditions between each 4-plot trial run varied from sunny with minimal cloud and dry to 100% cloud and wet ground. No rain occurred during the trials. The weight of Clay Soil Vent varied from 205 g to 583 g, Sandy Soil Vent varied from 224 g to 464 g, Clay Soil Full varied from 421 g to 661 g and Sandy Soil Full varied from 250 g to 575 g. The weight of individual great crested newts varied between 4.0 g and 12.0 g, at a length of 40–90 mm.

Overall detection time ranged between 2 s and 180 s. Out of the 128 trials, the dog/handler team exhibited 88% successfully (blank plots included). False indications (12% of trials) were significantly most commonly associated with blank plots ($n = 7$; $\chi^2 = 4.25$, d.f. = 1, $p = 0.04$), however also occurring in Sandy Soil Full ($n = 4$), Sandy Soil Vent ($n = 2$), Clay Soil Vent ($n = 1$) and Clay Soil Full ($n = 1$) treatments where *T. cristatus* were present. Within occupied plots, treatment had no effect on the frequency of false indications (Table 2). Locations of false indications did not appear to be influenced by wind speed or direction, location within the plot or location of previously placed *T. cristatus*. Six of the 15 false indications took place on the first day of the trials (19/08/22).

Times for successful detections ranged between 7 s and 180 s for the Sandy Soil Full treatment, between 2 s and 50 s for Clay Soil Full, between 2 s and 62 s for Sandy Soil Vent and between 4 s and 103 s for Clay Soil Vent (Fig 4). Detection time was significantly greater in the Sandy Soil Full treatment compared with all other treatments (GLMM: (log) effect size = 0.59 ± 0.18, d.f. = 76.86, $t = 3.26$, p <0.01, Table 3). This was supported by *posthoc* analyses which revealed no significant differences between any of the other treatments (Table 4).

**Table 3. Outputs from a restricted maximum likelihood (REML) GLMM of the effect of soil treatment on detection time, with air temperature specified as a random effect (*var.* = 0.19 ± 0.44).** Results are given on the log scale.

| Treatment | Estimate ± SD | df | t | p |
| --- | --- | --- | --- | --- |
| Clay Soil Full; Intercept | 2.48 ± 0.16 | 61.04 | 15.74 | <0.001 |
| Clay Soil Vent | -0.14 ± 0.18 | 77.97 | -0.79 | 0.43 |
| Sandy Soil Full | 0.59 ± 0.18 | 76.86 | 3.26 | <0.01 |
| Sandy Soil Vent | 0.07 ± 0.18 | 74.96 | 0.37 | 0.71 |

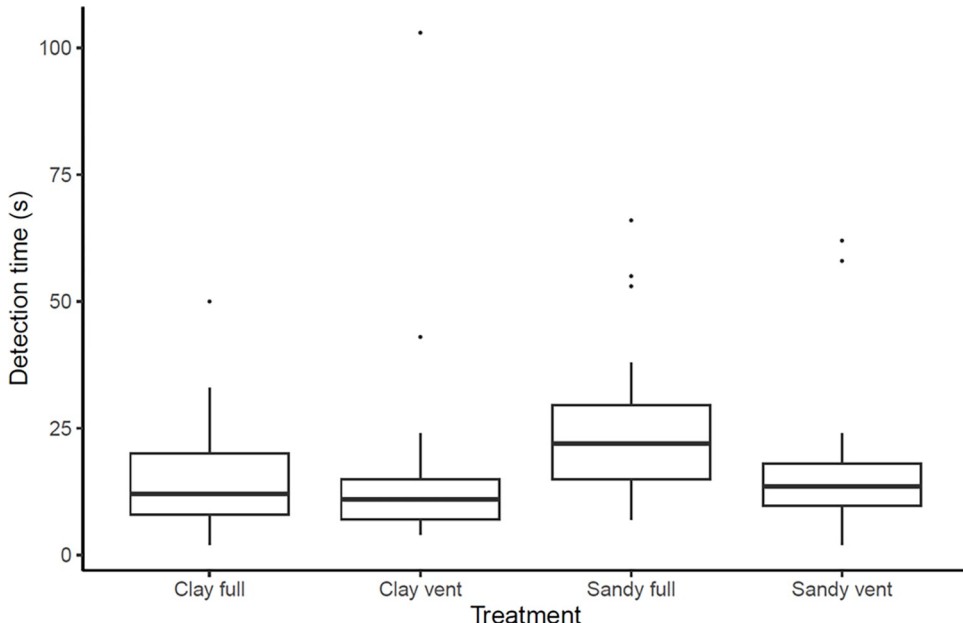

**Fig 4. Boxplots displaying detection times of *T. cristatus* in different soil treatment types by a detection dog.** Two outliers (180 s for Sandy Soil Full, and 103 s for Clay Soil Vent) are not shown to improve the graphical presentation. For more details see text.

**Table 4. Pairwise comparisons of differences in detection time across treatments.** Post-hoc analyses were performed on the estimated marginal means computed from GLMMs with a log-link. Results are given on the log scale. P-values were adjusted using the Tukey method.

| Contrast | Estimate ± SD | df | t ratio | p |
|---|---|---|---|---|
| Clay Soil Full–Clay Soil Vent | 0.14 ± 0.18 | 77.9 | 0.78 | 0.86 |
| Clay Soil Full—Sandy Soil Full | -0.59 ± 0.18 | 76.8 | -3.24 | <0.01 |
| Clay Soil Full—Sandy Soil Vent | -0.07 ± 0.18 | 74.9 | -0.37 | 0.98 |
| Clay Soil Vent—Sandy Soil Full | -0.74 ± 0.19 | 83.0 | -3.88 | <0.01 |
| Clay Soil Vent—Sandy Soil Vent | -0.21 ± 0.18 | 75.3 | -1.18 | 0.64 |
| Sandy Soil Full—Sandy Soil Vent | 0.53 ± 0.19 | 78.5 | 2.86 | <0.05 |

The variance explained by the random effect of air temperature was 0.19 ± 0.44; indicating a positive relationship between detection time and air temperature (see S2 Fig)).

## Discussion

*T. cristatus* has been detected successfully on the soil surface by trained detection dogs with a distance perception of approximately 20 cm [35]. The distance at which the scent is perceived can however be influenced by abiotic (e.g. air temperature, habitat type) and biotic factors (e.g. size and sex, [35]). *T.cristatus* inhabits subterranean shelters in channel-like refuges such as mammal burrows and rocky crevices [10, 11, 42, see also 40 for a closely related species]. We therefore investigated whether distance, when channelled, and soil interference have an influence on detectability as well as accuracy and speed of detection. Detection dogs have successfully been trained to locate different types of odours such as human remains or oil through different types of soil up to 5 m in depth [43, 44]. To the best of our knowledge this is the first wildlife detection dog study which explicitly investigates the influence of two contrasting soil

types on the ability to locate urodele amphibians in subterraneous shelters. The most similar study to date [45] assessed whether a detection dog can locate the giant bullfrog *Pyxicephalus adspersus*. Although *P. adspersus* is considerably larger than *T. cristatus*, weighing around 2 kg, this species is also subterraneous, found in around 1 m depth burrows. The study looked at whether the detection dog team could locate *P. adspersus* in three phases: 1) at the surface in containers 2) diluted target scent and 3) in simulated natural environments such as burrows. The detection dog team was successful at locating the odour in all three phases of the experiment and successfully detected bullfrogs in burrows in the wild.

The channelled distance perception trials concluded that *T. cristatus* can be detected from up to a minimum of 2 m at temperatures of 21–26˚C without substrate interference. These findings differ from a previous study which found that distance perception of *T. cristatus* was maximally 20 cm above ground when the scent was not channelled [35]. The presence of *T. cristatus* within a structure such as a mammal burrow could therefore funnel the scent at a greater distance. The pipes used for the experiment were however positioned in direct sunlight and made of heat absorbing material (black plastic), potentially increasing the distance of odour dispersion along the pipe [46]. The results may therefore not be representative of below ground conditions, as subterraneous burrows have a different microclimate [47]. Repeat trials should therefore position pipes below ground, similar to the soil interference trials and [44]. Burrow microclimates (temperature and humidity) as well as gas diffusion is further influenced by vegetation cover, soil type, depth of burrow, length, diameter and shape as well as burrow architecture [47]. Future studies should consider these factors and how they may influence the dispersal of scent within a burrow, in addition to a setup which enables a more rigorous statistical analysis. The two false indications were exhibited at the beginning of the trials, which may have been due to wind moving scent along the rig causing an early indication (as also observed by [45]), and repeating trials in a sheltered environment would control abiotic influences. Manipulating air temperature ranges would also determine whether detection probabilities at given burrow architectures differ for example between warm summer months and during hibernation.

The soil interference trials concluded that the detection dog/handler team achieved a 88% success rate to locate individual *T. cristatus* at 200 mm depth in different soil treatment types including blank treatments. False indications (12%) were largely exhibited during the first set of trials. The handler noted that the detection dog was in an aroused and distracted state at the start of the trials and therefore modified the length and type of play, including calming prior to the search to keep the dog focused. This resulted in more focused searches and a reduction in false indications for the remainder of the trials. False indications were also noted on empty controls, when the handler did not call "blank" immediately after the dog had searched every container. False indications following a blank search is likely associated to the detection dog failing to find a target and take the chance of getting a reward by displaying a false positive indication on a blank pipe [39]. The handler became more confident at reading the detection dog behaviour as the trials progressed, therefore calling "blank" earlier during the trial. False indications were unrelated to the position of *T. cristatus* in previous rounds, rendering indications on residual scent unlikely.

The presence of vents yielded a faster response and less false positive indications in comparison to full treatments irrespective of soil type. When *T. cristatus* was placed in Clay Soil Full, detection speed and accuracy was higher in comparison to Sandy Soil Full. These results contrast with findings derived from studies undertaken on detection dogs trained to find human remains, which were able to locate their target at a shorter time in sandy soil in comparison to clay [43]. Human remains likely emit a stronger scent profile including decomposing gasses which travel through porous media such as sand, reaching the surface more freely [48–50].

Alive *T. cristatus*, on the other hand, likely have a lower scent profile and generally rely on water to transport pheromones during their aquatic breeding phase [51, 52]. Clay soil generally contains a larger amount of water than sandy soil and may therefore transport the scent of amphibians more freely in comparison to sandy soil [46]. Alternatively, the scent from the study individuals may be more tightly retained within the moisture of clay soil in comparison to sandy soil where the scent may evaporate more readily [46]. That the odour penetrated less through sandy soil is also corroborated by our finding that, aside of the empty controls, the largest number of false indications and longest time to locate the correct pipe took place for the Sandy Soil Full treatment. The target odour and how it reacts with the substrate may therefore influence the detectability of *T. cristatus* during surveys. The detection dogs' olfaction physiology may have a further influence on how information is absorbed. Olfaction occurs when a dog draws in odorant-laden air into its naval cavity by sniffing, thus transporting odorant molecules from the external environment to olfactory receptor neurons in the sensory region of the nose [53]. Highly soluble odorants are generally deposited in the front of the olfactory recess along the dorsal meatus and nasal septum, whereas insoluble odorants are deposited throughout the entire olfactory recess, thus decreasing odorant loss through absorption on non-olfactory surfaces. Due to the higher moisture content, *T cristatus* odorant may be more soluble in clay soil in comparison to sandy soil.

Sandy soil moisture levels were below 10% and clay soil moisture levels were around 30%. Dry soils typically absorb large amounts of Volatile Organic Compounds (VOCs) on soil particle surfaces and leave a residue that reduces available scent levels [46]. As the water content increases, water replaces the VOCs absorbed on the soil particle surfaces and releases them for possible transport to the surface and atmosphere. High water content can however have a negative impact on scent movement by replacing gas phase transport with water phase transport [46]. Therefore, surveys of subterranean *T. cristatus* during heavy downpours may yield poor results. Optimal detection conditions of *T. cristatus* in subterraneous shelters may therefore be following periods of light rain and when dew is present on the ground. Further studies should manipulate moisture content in both sandy and clay soil to investigate whether this has an influence on the detectability of *T. cristatus*.

In line with [35], air temperature was negatively related to the speed of *T. cristatus* detection in all four treatment types and was inversely related to air humidity. Nevertheless, air humidity was unrelated to detection time, contradicting other similar studies [35, 54–56]. Whereas [57] also found no correlation between air humidity and detection probability of pointing dogs locating game birds, this study revealed and influence of solar radiation, parameters which are as yet unaccounted for in *T. cristatus* detection dog studies. Abiotic factors such as wind speed and wind direction did not appear to have an influence on the detection of *T. cristatus* when located in artificial subterraneous refuge. This may be due to the scent being held within the substrate in comparison to when the scent is within the open environment. Wind speed and direction of the wind was also taken at head height rather than close to the ground.

We found no relationship between detection dog performance and sex or size of *T. cristatus*. This contradicts another study [35] which found that males had a higher detectability than females for individuals retrieved during their aquatic breeding period and located on the soil surface. Further studies investigating distance perception and detection probability during different stages of the biphasic lifecyle of *T. cristatus* are recommended.

For our experiments, the detection dog underwent specific training prior to the assessment to learn to indicate to the handler at a distance from the odour without getting within close proximity. This is an important aspect when training dogs to detect wildlife species such as *T. cristatus*, which are not always directly accessible at the surface [10, 11]. Training of the handler is also important to ensure they are able to detect behavioural changes when diluted

sources of the scent are encountered [15, see also 34, 43]. It is however important to ensure that training with diluted scent solutions does not result in indications on residual scent, as during mitigation measures it is vital to locate the individuals only in occupied and not in vacated subterraneous shelters that may have retained the scent. Future studies should determine to what extent detection dogs have the ability to distinguish between residual scent and scent of individuals located at distance.

Locating *T. cristatus* during their subterranean phase with the use of detection dogs can provide novel insights into their terrestrial habitat preferences, as well as locating them for translocations for example as legally required prior to construction activities. This study highlights how environmental factors such as temperature and soil type can influence the speed and accuracy of a detection dog to locate *T. cristatus* in subterraneous shelters. These findings are important when working operationally, as the identified factors likely impede or enhance detection probability. A further important consideration is that long periods of training are required for *T. cristatus* detection dog teams due to the complexity of the target odour, thus constraining the rapid deployment of effective dog/handler teams for given tasks [14, 15]. This study highlights the importance of an awareness of the environmental surroundings by handlers, as abiotic factors can influence the diffusion of scent molecules released from the target odour. Effective handlers should be able to adapt suitable handling techniques and methodologies to enhance the capabilities of detection dogs to increase the probability of successful finds.

## Supporting information

**S1 Fig. T.cristatus being measured from nose to cloaca.**
(PDF)

**S2 Fig. The relationship between (log) detection time and air temperature with predicted values + SE from a gamma GLM (log effect size = 0.59 ± 0.09, t = 6.34, p <0.05).**
(PDF)

**S1 Table. Results of the weather conditions during the channelled distance perception trials.**
(PDF)

## Acknowledgments

We are grateful to Hanson Aggregates, Chipping Sodbury, for allowing us to capture *T. cristatus* from their receptor site and carry out research trials on their land. We are also grateful to research assistants; Jamie Mitchell, Willow West, Gemma Pyke, Rebecca Howell and George Gregory, and to Natural England for issuing the licences which made this research possible (licence numbers 2019-39743-SCI-SCI; 2020-45697-SCI-SCI; 2021-51164-SCI-SCI).

## Author Contributions

**Data curation:** Nicola Jayne Glover.

**Formal analysis:** Nicola Jayne Glover, Amy Leedale.

**Investigation:** Nicola Jayne Glover.

**Methodology:** Nicola Jayne Glover, Louise Elizabeth Wilson.

**Project administration:** Nicola Jayne Glover.

**Supervision:** Louise Elizabeth Wilson, Robert Jehle.

Writing – **original draft:** Nicola Jayne Glover.

Writing – **review & editing:** Amy Leedale, Robert Jehle.

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
