## [Decision Letter · Decision Letter 0]

2 Oct 2022

PONE-D-22-22016An experimental assessment of detection dog ability to locate great crested newts (Triturus cristatus) at distance and through soilPLOS ONE

Dear Dr. Glover,

Thank you for submitting your manuscript to PLOS ONE. After careful consideration, we feel that it has merit but does not fully meet PLOS ONE’s publication criteria as it currently stands. Therefore, we invite you to submit a revised version of the manuscript that addresses the points raised during the review process. Both reviewers considered that your manuscript has merit and is potentially of great interest to others working in the field. However, both also raised significant questions regarding the statistical handling of the results which must be addressed if it is to go forward for publication and listed numerous minor issues of terminology and presentation which would greatly enhance comprehensibility if they were addressed. Note that one reviewer also requested the inclusion of supplementary information.

We look forward to receiving your revised manuscript.

Kind regards,

Christopher Walton, Ph.D.

Academic Editor

PLOS ONE

Journal Requirements:

Reviewers' comments:

Reviewer's Responses to Questions

**Comments to the Author**

1. Is the manuscript technically sound, and do the data support the conclusions?

Reviewer #1: Partly

Reviewer #2: Partly

2. Has the statistical analysis been performed appropriately and rigorously? 

Reviewer #1: No

Reviewer #2: I Don't Know

3. Have the authors made all data underlying the findings in their manuscript fully available?

Reviewer #1: No

Reviewer #2: No

4. Is the manuscript presented in an intelligible fashion and written in standard English?

Reviewer #1: Yes

Reviewer #2: Yes

5. Review Comments to the Author

Reviewer #1: I had the pleasure to review the article "An experimental assessment of detection dog ability to locate great crested newts

(Triturus cristatus) at distance and through soil", which descripes two new testing procedures for a GCN detection dog. This topic is highly relevant due to the global amphibian decline and the limited knowledge we have about the newts' terrestrial phase.

The manuscript is very well written and generally follows a red line. I would like to congratulate the authors to their very creative testing procedures and apparatuses. This is a great accomplishment and will shed light into the secret of newt detection dogs, if not amphibian detection dogs in general. That said, I have a long list of additional information that is necessary to follow the setups entirely and procedures, please see my attached review for details. Regrettably, test 1 only had 2 repetitions for 8 conditions, making comprehensive statistics almost impossible. In contrast, the concept of test 2 seems well thought-through with 4 conditions and 128 repetitions. This gives room for rather sound statistical analyses. However, statistical analyses done afterwards are rather weak. A pairwise comparison of findings that at the same time may (or may not) depend on so many other factors contradicts the test assumptions. Please see my suggestions in the Methods section. Expanding the analyses to the proposed ones would empwer the findings of the article dramatically and would give much more weight to them, especially when comparing them to other studies.

I have some concerns regarding some terms used which can easily be adapted by the authors. This also includes more precise definitions of the weather parameters and a more careful use of the term "detection distance". Nevertheless, these are rather minor points.

Please see my comprehensive review attached.

Reviewer #2: General comments:

The authors present an interesting study on the ability of a detection dog to locate an urodel amphibian in different soil types and at different distances. As an herpetologist and a dog owner myself, I find this emerging practice very interesting and with great potential and wide range of possible applications in the future.

I find the paper generally well written, with clear experimental methods, however, I have some concerns about data analyses.

My main concern is about the analysis of soil data, which are not clear to me. How was the relationship between environmental/experimental variables and time to detection tested? Just with Spearman’s r test (line 206)? In this case, analyses should be performed with glm relating time to detection ~ abiotic + biotic factors. Just using pairwise Spearman’s r means that the authors are completely ignoring joint effect/marginal responses of dependent variables.

I don’t know whether the sample size is enough to include an additional variable; in case, you can test the effect of progressive trial number of detection performance.

I would avoid to state “data not show” and add supplementary instead (I agree that “data not shown” can be fine for the video footage mentioned at line 304).

I can’t find raw data for the soil experiment, even if they should be available given the data availability statement. Maybe they can be added as SI?

Specific comments:

Line 80: “a/one” detection dog, maybe?

L100: “derived” do the authors mean they were captured in a quarry?

L106-107: I would move “T. cristatus” eralier in the sentence, before involing “scent”.

L108-110: Where all the trials conducted in a single day? Is it the same of one of the soil trials or another day?

L118: the comma after [33] should be a dot.

L118-119: Please, add more information on this issue.

L119-120: I’d like more detail also here to understand why water spray is necessary.

L137: OS is extended here and abbreviated at line 101, it should be the opposite.

L229-230: It is not clear to me the meaning of Wet-Wet+ and Dry-Dry+.

L258: Strange sentence construction.

L264-265: “a single wildlife detection dog only”, I would remove “only”.

L271: I would change “mortality” with “enhanched/augmetned/increase in mortality” since mortality in general is a natural phenomenon.

L271-273: “and due to limited battery duration at required small transmitter sizes is presently unable to locate newts over periods longer than a few weeks after breeding” I think something is missing in this sentence.

L275-276: Something is wrong with the punctation, or some word is missing.

L276: I would remove “for example”, same at line 284, 287, 317.

L279: And also useful for monitoring purposes I suppose.

L286: “individuals of”

L288-290: Please rephrase and try to merge the two exapmples.

L294: Maybe better “wide availability”.

L299-301: Strange construction, please rephrase.

L324: Here and above, why “Sandy Soil Full” is capitalized?

L326: And another main point could be to use more dogs to assess the variability of detection dogs’ ability to locate newts.

L478: “B” should not be capitalized.

6. PLOS authors have the option to publish the peer review history of their article (what does this mean?). If published, this will include your full peer review and any attached files.

Reviewer #1: No

Reviewer #2: No

---

## [Author Response · Author response to Decision Letter 0]

16 Feb 2023

Comments by Reviewer 1:

Introduction

The introduction is very well written and overall follows a logical flow. My only suggestion would be to start from the ecological point of view (i.e., line 46 – maybe not with a “To”), and once the terrestrial methods for surveying newts are brought up (currently ending line 76), come to the method of detection dogs. This would prevent from this sharp cut that currently exists between paragraph 1 and 2.

We are grateful for this constructive comment. We have re-ordered the Introduction accordingly, starting with the ‘ecological view’ and study species before presenting a section on detection dogs. 

line 42: references 6-9: three of the references refer to carcass monitoring. Although this is true, it seems a little less variable to generalize that dogs are generally 4-12 times better than humans. I would suggest that the authors also include papers on other species or their traces.

We added references 22-24 of other example target odours as suggested. 

line 49: please give some references on those “easier” sampling methods

Additional references (2 and 3) on amphibian sampling methods are now provided.

line 51: again the addition of some typical sampling methods and their efficiency would help the readers

Further general texts on sampling methods (references 4 and 5) are now provided. 

line 62: it is also an annex 4 species of the FFH directive

This information has been added (lines 47-48). 

Methods

Generally, I would like to congratulate the authors to their very creative testing procedures and apparatuses. This is a great accomplishment and will shed light into the secret of newt detection dogs, if not amphibian detection dogs in general. That said, I have a long list of additional information that is necessary to follow the setups entirely and procedures.

The description of test 1 was quite good while there were several logical flaws in the description of test 2 (see below). I am also missing some information on how habituated the dog was to the apparatuses and the setups, since mistakes often occurred in the first trials. 

In contrast to the writing, the concept of test 2 seems well thought-through with 4 conditions and 128 repetitions. This gives room for rather sound statistical analyses. However, in test 1 there were only 2 repetitions for 8 conditions making further analyses almost impossible. Despite the finding is interesting, the concept is also biased towards learning by the dog as the length used per trial was not randomly chosen but increased systematically, limiting the potential to generalize these findings. This should be explained and taken up in the discussion.

We are grateful for these constructive, detailed and encouraging comments, and have indeed taken them up throughout the revised manuscript. 

We are fully aware that the setup of “test 1” (the channelled distance perception trials) is characterised by an ascending (and therefore non-random) order of pipe length, limiting the scope for detailed quantitative analyses. However, the main purpose of this experiment was to demonstrate the ability of our dog to detect the target species at distances of up to 2 m, which was unambiguously demonstrated in the conducted trials (the longest pipes never resulted in non-detection). More rigorous statistical analyses at a more randomised setup would indeed be required at extended pipe lengths which gradually become a limiting factor for detection, but this was beyond the scope of our experiment. 

Finally, despite test 2 is following a good concept, statistical analyses done afterwards are rather weak. A pairwise comparison of findings that at the same time may (or may not) depend on so many other factors contradicts the test assumptions. I would suggest a linear regression (linear model or an anova) for these analyses, with the detection time as a response and properties of the individuals (individual measurements, sex) and environmental conditions (weather in this case, as long as weather parameters are not correlated, of course) as explanatory variables. After reading the results, I also figured out that the dog did a lot of false-positive indications. I would thus use a binary coding of whether the dog was able to detect the newt at the first attempt (1) or not (0), neglecting whether or not the dog would have found the individual in any following attempt (because the likelihood of detection is different from the first attempt). This binary coding (i.e., detection probability) would then be used as a response in a glm with Binomial error distribution and all the above-mentioned individual and environmental parameters as explanatory variables. 

We have taken these suggestions on board, and present a new set of data analyses based on GLMMs (see Methods and Results sections). 

line 87: as experiments were conducted in different year, it may be better to give the year of birth or a reference when the dog was five years old (at the beginning? last test?)

A birth date of the detection dog is now provided (see lines 93 and 94).

lines 89-90: what is meant by “annual assessments”? How is the dog assessed and what for? Is there any standard protocol the authors could refer to?

More details are now provided in lines 94-102. The trainer shadows the detection dog team during operational searches and sets up controlled trials to assess working capabilities. There is no standard protocol that can be referred to, as the assessments were developed and designed by one of the authors at Conservation K9 Consultancy. 

line 90: likewise, please define “suitable operating standards”

This section has been rephrased (see above). 

line 92: this is a good example of how such an assessment can be cited

We are grateful for this comment.

line 92-93: “Freya is able to distinguish between the target species and all other common amphibians native to the UK” – after having a look at reference 31, it is not clear to me how the authors define “is able to…”. Would the authors consider giving the “score” for Freya? According to ref. 31, it seems that 80% GCN detected and no more than 2 false positives is a “pass”, but 50% or lower GCN detection and 4 or more false positives are a “fail”. That makes it hard for the reader to evaluate the dog performance. 

More details are now provided in lines 98-103, to clarify that the detection dog achieved a success rate of over 80% and therefore passed the internal assessment criteria.

line 100-110: I would like to congratulate the authors to such a very smart setup. This is a great idea. I have a few questions: Would the authors consider also sharing a picture of the bottom of a pipe (description lines 103-106), maybe as an inlay to Fig. 1? 

Thank you for the compliment! A picture of the bottom of the pipe has been included as an inlay to Figure 1. 

Why was the test restricted to two runs per length (which results in very low statistical power)? Was that because 16 individuals were available, and no individual should be used more than once? 

The number of trials was limited by the licence issued by our governing body (Natural England), which required to use each individual great crested newt only once to minimise stress. This is now clarified in the manuscript (line 120-122).

And last, I have some concerns with the “successively increasing the length” as this could result in a learning effect, while on operative searches it is more likely that the dog encounters various lengths randomly. Results may be different if a dog is tested with a successively changing situation (which would rather be the approach of a learning protocol than for a test) in comparison to a randomly changing situation (a short pipe run followed by a long and then a medium pipe run etc.). 

See our general comments above – we are aware of this limitation, which is now considered in the manuscript (e.g. lines 247-250). 

Last, was the position of the board always the same in respect to the pipe length that exceeded the board? So that for the dog the setup always looked identical?

Yes, the set-up always looked identical – this is now clarified in line 120. 

line 114: please specify “blind” to “double blind”

Double blind refers to when the dog/handler team and observer are out of sight when the testers set up each run. Alternatively, a blind search is when only the dog handler team are unable to see the test setup, but the observer is aware of the location of the target odour. These types of setups are carried out to avoid unconscious signalling from the handler or observer/tester to the detection dog, and is now clarified in the manuscript (lines 123-126, lines 214-216).

line 115: please insert: “from the handler” or the observer/tester “to the detection dog” (see below for wording confusion)

The wording has been amended.

line 123: “one additional repeat” – backwards or starting again?

This information has been added (the run started again, see line 140-141).

lines 127-129: please specify the parameters taken during the test, e.g. what were the weather conditions and how were they defined, how was temperature / wind speed / humidity measured, what categories existed for the behavior of the dog and handler and how were they defined?

The information is now added (lines 145-160).

line 130: the “tester” is the same person as the “observer”? Then, I would suggest to use a unique name. If they are not the same, I did not understand the tasks of both so far. Please also verify again which of these persons was placing the newt and then went out of sight of the handler, and whether the third person (if there was any) observed the whole procedure. If so, the test is no longer considered as double-blind but single blind.

We apologise for the confusion we have caused. The tester set up the trials and stood out of view from the dog/handler team and observer. The observer did not know the location of the target odour and noted behavioural observations during the assessments. The tester in the soil trial was also an observer therefore the trial was undertaken blind (not double blind). We have amended the terminology to “assessor” and provide more details in the manuscript throughout. 

line 156: the setup is not visible in Fig. 2a. Would the authors please consider adding a scheme of the setup?

A scheme of the setup is now provided (Figure 2a). 

line 161: this may be a funny question, but how is the 68 mm pipe attached to a 75 mm pipe adaptor? It may just be my version of the pdf, but Fig. 2b is a little too blurry to see it

We are grateful for this comment, and apologise for an error. The pipe socket (the compartment where the experimental individual is housed) has a 61mm diameter. The 20 cm pipe that slots over the top is 68 mm in diameter. See lines 187-199. 

line 164: this sentence gave me a hard time to understand the setup. I suppose that the 1x1 cm holes are supposed to be the mesh size? And the mesh has been rolled to a diameter of 3.75 cm? Again, a drawn scheme would be helpful here.

We apologise for these omissions, and have now added all measurements. The fibreglass mesh that was placed between the compartment and the pipe comprised 4 x 4 mm gaps (lines 191-192 and Figure 2a). The vented treatments comprised a plastic-coated wire mesh with 4 x 4 mm gaps. The mesh was cut to 200 mm length to sit within the 200 mm long pipe. . The mesh was rolled up into a cylindrical shape and secured with cable ties with a diameter of approximately 25 mm to represent small mammal burrows (a common subterraneous refuge of T. cristatus (195-199 Figure 2b).

lines 166-170: this part needs some more details to understand the whole setup. As far as I understood it, 8 pipes per plot existed, one was holding a newt (or more?)? And all other soil types could exist? So two pipes per soil type in each plot? Please specify.

More details are now provided in lines 204-213. Two pipes of every treatment type were present in each plot with one soil type (e.g clay) located to the left of the plot and the other soil type (e.g. sandy) was positioned to the right, alternating the location of each soil type between runs. On each side, vented and full treatments were positioned alternatively within each plot. Three out of the four plots contained a single T. cristatus, with one plot serving as a control. Across four trials and four plots, the location of three T. cristatus was allocated to each of the four treatments at random locations, using 12 T. cristatus in total. 

line 173: again this my seem to be a funny question, but if the pipe was 20 cm height and the adaptor was 5 cm height, but the hole was just 22 cm depth, the pipe must protrude 3 cm above surface. Was that on purpose? How does that affect the setup?

We are grateful for this comment, and apologise for a wrong measurement. The pipes sat flush with the ground (Figure 2b), and the numbers are now corrected. 

lines 175-177: the information about the weighting of the crested newt should go along with line 135 where the newts are mentioned.

The section has been amended accordingly.

lines 178-179: again, please define the parameters properly. Also clarify why you added cloud cover to your set of parameters

This information has been added (lines 145-151). 

line 179: is that soil moisture measurement different from the procedure described above, i.e. has it been repeated during the tests?

Soil moisture measurements were taken once before each trial took place to minimise disturbance. It took a maximum of 10 minutes to conduct searches across Plots 1-4, rendering it unlikely for moisture and temperature to change drastically (see lines 220-223 in the revised manuscript). 

line 183: this sounds very negative and pessimistic about the dog. I am sure also the true positives were recorded. � Likewise, the number of misses (false negatives) were likely also recorded (or would be, if there were any). I would appreciate a more complete picture in that sentence.

This sentence has been rephrased (lines 248-250). As measures of performance by the dog/handler team, numbers and locations of false negative, false positive, true negative and true positive indications were noted.

line 183-184: again I am getting confused with the tasks for tester and observer. Please clarify their tasks and who was present at the plot and when. Also, it should be mentioned that the test was conducted single-blind (despite you described it in the text that the handler did not know the location, readers my search for blindness in your article).

The text and terminology has been amended accordingly (see also above). 

line 186: the dog “undertook a free search” – but actually, Fig. 2a suggests that the dog was working on a long towing leash; please clarify; another upcoming question: how did the dog search? did she deliberately check the 8 tubes? did the handler follow the dog or stood still at the entrance of the plot?

The detection dog was only on a lead when being positioned at the front of the plot, and a different photo is now shown to avoid any confusion (see Figure 2c). 

line 186: “dog indicated correctly” – before the handler confirmed with the tester, it is not necessarily correct. Rewrite sentence in more logical order, e.g., dog indicated – handler confirmed with tester – if correct, play session

The sentence has been amended accordingly. 

line 187-190: this sentence seems to be a bit miss-placed and would be more in logical flow after line 191. I suggest to change this.

Agreed and amended.

line 188: If I understood the setup correctly, there was always one empty plot among the four plots. Was the handler able to stop the search before the 180 sec and state that she believes that nothing is here? please clarify.

Yes, the handler would call “blank” early if she believed there was no great crested newt present due to no change in dog’s behaviour. This is now clarified in line 232-234.

line 191: and also the number of attempts as well as the time until the dog would find the newt was noted, I guess? please specify

This information is now provided.

line 194: I think I am not yet familiar with what the authors determine as “pipe” and “tube”. While line 192 suggests that new tubes were used during each trial, line 194 states that all pipes were moved to new positions. Please clarify this issue and specify the terms used.

Amended to ‘pipes’.

line 197: how were the 30 newts distributed among the 128 trials (even though some were without newts)? was the selection completely random or pseudo-random (i.e. allowing for a similar number of trials per newt)?

Great crested newts were separated for each trial to avoid their re-use during the same or next trial day to reduce potential stress. Individual newts were taken at random from the tanks. This is now clarified in lines 244-246.

 On another note, there is again a wording issue with “trial”, which the authors sometimes use for all four plots together and sometimes for a single plot (at least this is how I understand the 128 trials). Maybe the authors could think of a more appropriate term, or specify (as done sometimes but not throughout the manuscript) as 4-plot and single-plot trials

The terminology has been amended to 4-plot and single plot trials as suggested.

statistical analyses: the statistical part should follow the same order as the field methods part. It would thus be better to start with statistics of test 1; if no statistics for test 1 were done, please at least describe some potential measures for this test; unfortunately, more repetitions would have been helpful for a more comprehensive understanding

The statistical approach of the initial manuscript version has been replaced by a GLMM, and the entire manuscript section was re-written accordingly. The lack of a testing framework for “test 1” is explained in lines 253-255. 

line 200: what “data” are the authors talking about? To understand this part, a general context on what the authors aim to analyse would help. For example, differences among detection rates or detection times regarding the soil type etc. Were the detection times checked for normality? Or what else?

See above – a new statistical approach has been adopted and the section re-written.

line 201: I would suggest to delete the part: “and box plots were used to visualize the data” as this is obvious information

This section has been removed.

line 203: medians of what?

This section has been removed.

line 204: differences of what?

This section has been removed.

line 205: aren’t vents/ no vents and sol/clay the four treatments? What is the difference to the KW-Test?

This section (and test) has been removed.

line 206: how do the authors define detection rates? which parameters belong to the abiotic and biotic parameters? The last part can be added to my question above about a more detailed parameter definition

The new statistical model is now described in lines 256-271, and the underlying abiotic parameters are described in lines 145-151.

Results

Results are well written but hard to generalize in their current state. Please consider a more sophisticated statistical approach as the one suggested in the Methods part. Also, original data are missing and the authors should consider uploading them to a repository when publishing the article. 

The statistical analyses of the initial manuscript version have indeed become by a GLMM and surrounding procedures. The raw data have been made available on Dryad .

Table 1: I forgot to ask: how long was the dog trained on the setup before? How used was she to search on it? Again, due to the successive increasing of the pipe, a learning effect cannot be excluded here. 

The detection dog was trained for two weeks on the setup prior to assessment and had therefore become habituated to the process. This is now included in the Method section (lines 158-160).

line 228: soil surface or somewhere within the soil?

This information is now provided (5 cm within the soil, see line 221). 

line 227-232: despite this information is very interesting, it does not tell much without the corresponding outcome. A summary with the outcome data would help, be it detection probability or time to detection

We would prefer to retain this information in the revised manuscript version, as the newly adopted statistical analyses link it more closely to the outcome of the conducted experiments. The outcome data are now provided via Dryad. 

line 237: more than 20% indication in blank plots seems quite high and would be difficult in real deployments when only retreats can be found but not the newts themselves. Any ideas why this happened? Could residual odor play a role? Again, also the search strategy would be interesting to know, i.e. whether the dog systematically checked the tubes (by herself or with a handler following).

The increased (but overall still low) frequency of false indications is now described in detail in lines 307-314 (Results) and lines 365-378 (Discussion). The search methodology is explained in lines 226-230. 

line 240-241: this statement must be underlined with statistics, otherwise it is too blurry. See my suggestions about a potential statistical analysis in the Method Section

The section has been re-written together with re-analyses of the data. 

line 242: as for test 1, I am wondering how familiar the dog was to this setup. 

The dog experienced three weeks of training prior to assessment – this is now clarified in the text (lines 247-248). 

Figure 3: please add pairwise significant differences to the plot

The data became re-analysed, and the plot became modified accordingly. 

lines 244-251: directions (effect sizes) are entirely missing. If things are significantly different, which one was faster/slower?

The section has been re-written together with re-analyses of the data. 

lines 251-253: again, please underline the statements with statistical results

The section has been re-written together with re-analyses of the data. 

Discussion

General: The discussion is very well written and follows a red line. Unfortunately, discussions of own results are rather short (despite including very nice parts, see below). If the authors would consider a full analysis on detection at first attempt and detection time depending on all parameters measured (as suggested above), results would be much more comprehensive and could directly be compared to more findings from others. In line with that, the authors claimed that they did not detect any differences between sexes or with size, which contradicts recent findings from a different detection dog on GCN (Grimm-Seyfarth 2022). Having a look at the whole models (sometimes parameter influences can only be detected when controlling for other, maybe more important parameters, such as potentially the soil type in this case) and comparing results would be extremely interesting. 

A conclusion is missing.

We are grateful for these constructive comments, and now present an extended and modified Discussion. We particularly integrate the additional reference (Grimm-Seyfarth 2022, which was not yet available at the time of writing of the first manuscript version) into the interpretation of our findings. 

line 265: and limited to a single amphibian species

This section has been amended.

line 281: careful, the authors did not evaluate what is commonly referred to as detection distance. I typical detection distance would be when an individual / trace is placed and the dog would wind its head and change direction at xx m distance and then move straight to the target. The authors however used a lineup with different pipe lengths, were the smell would not be distributed through the environment but channeled in a pipe, likely coming close to a chimney effect (Osterkamp 2000). The amount of smell through such a pipe is much higher than would be on the surface. Nevertheless I do understand the applicability of these findings when newts are hidden in the ground. Your findings tell us that the dog would find newts in 2 m deep mammal burrows (if they would be straight etc.), which is a very important finding. However, I would strongly recommend that the authors use a proper term for it to not confuse it with typical detection distances.

We agree with this comment. Distance perception has been evaluated through the use of pipes which could have potentially increased distance of scent flow. We have generally re-named these experiments as “channelled distance perception” throughout the manuscript, and cover this issue in the Discussion (lines 342-348). 

line 288: careful, reference 39 is talking about detection distances (measured accurately with GPS and in field-like setting), while the authors tested the distance channeled through a pipe. this is not really comparable and should be specified

This section has been amended.

line 290: reference 41 evaluated the distance from a defined transect line that a scat can be detected, thus also not really evaluating detection distance (which would properly be done like in ref. 39); please specify

We have removed the reference.

line 291: it would be helpful to not just know the temperature range exhibited during the test, but also whether hot temperatures occurred at the beginning or end, as temperature may influence the channeling effect of the scent

Temperature was high at the start of the test when false indications occurred (26 degrees) and dropped as trial went on (21 degrees). We have now included temperature in the statistical model, and present the data in Dryad. 

line 292: Grimm-Seyfarth 2022 investigated the effect of temperature (among others) on detection probability for T. cristatus, which apparently was also dependent on the habitat; and my suggestion would be to include temperature analyses on test 2 (see my suggestion in methods part)

Temperature is indeed now considered in the statistical model, suggesting a positive relationship with detection time (lines 325-327, and according sections in the Discussion). 

lines 298-308: these are important information, thanks for including them! I would appreciate to have some more information in the method section as requested above, as this may clarify some of my points risen above

The respective Methods sections became revised (see above). 

line 311: the studies by Matthew are indeed quite comparable to some points and I would suggest to draw some more direct comparisons regarding her and your results

We agree, and now present more detailed comparisons in the Discussion (lines 341-348).

312-326: this is a very nice part of the discussion! I would encourage the authors to include more of that. Comparisons among studies, even if different approaches have been used, and their findings are extremely helpful in this field.

We have included further comparisons as requested. 

References

I did not check all the references, but this came to me:

line 415: there seems to be a mistake with the title, see https://www.sciencedirect.com/science/article/pii/S0006320703002684?via%3Dihub

This has been corrected. 

Grimm-Seyfarth, A. 2022. Environmental and training factors affect canine detection probabilities for terrestrial newt surveys. Journal of Veterinary Behavior 57: 6-15.

Osterkamp, T. 2020. Detector dogs and scent movement. How weather, terrain, and vegetation influence search strategies. CRC Press, Boca Raton, USA.

We are grateful for these additional references. They are particularly useful and have been added to the manuscript.

 

Comments by Reviewer 2: 

The authors present an interesting study on the ability of a detection dog to locate an urodel amphibian in different soil types and at different distances. As an herpetologist and a dog owner myself, I find this emerging practice very interesting and with great potential and wide range of possible applications in the future.

I find the paper generally well written, with clear experimental methods, however, I have some concerns about data analyses.

My main concern is about the analysis of soil data, which are not clear to me. How was the relationship between environmental/experimental variables and time to detection tested? Just with Spearman’s r test (line 206)? In this case, analyses should be performed with glm relating time to detection ~ abiotic + biotic factors. Just using pairwise Spearman’s r means that the authors are completely ignoring joint effect/marginal responses of dependent variables.

We have taken these suggestions on board, and present a new set of data analyses based on GLMMs in the revised manuscript (see also a range of replies to queries by Reviewer 1 above). 

I don’t know whether the sample size is enough to include an additional variable; in case, you can test the effect of progressive trial number of detection performance.

We are grateful for this excellent suggestion, but would prefer to keep such analyses for a future manuscript to specifically focus on dog behaviour (including a comparison between different dogs). 

I would avoid to state “data not show” and add supplementary instead (I agree that “data not shown” can be fine for the video footage mentioned at line 304).

This section has been amended, and we now show all data in the Dryad repository. 

I can’t find raw data for the soil experiment, even if they should be available given the data availability statement. Maybe they can be added as SI?

All data are now made available via the Dryad repository.

Specific comments:

Line 80: “a/one” detection dog, maybe?

We have amended the phrasing to “a”. 

L100: “derived” do the authors mean they were captured in a quarry?

Yes – we have amended the word to “captured”.

L106-107: I would move “T. cristatus” eralier in the sentence, before involing “scent”.

Agreed – we have amended the sentence accordingly (lines 113-115). 

L108-110: Where all the trials conducted in a single day? Is it the same of one of the soil trials or another day?

Yes – Line 123 in the revised manuscript explains that “The experiment took place on the 20th August 2020.”

L118: the comma after [33] should be a dot.

This section became amended.

L118-119: Please, add more information on this issue.

More information has been added to the text (lines 129-135).

L119-120: I’d like more detail also here to understand why water spray is necessary.

We have now clarified that water is needed to keep the skin moist (line 135). 

L137: OS is extended here and abbreviated at line 101, it should be the opposite.

The section has been amended

L229-230: It is not clear to me the meaning of Wet-Wet+ and Dry-Dry+.

More details are now provided in lines 220-223. 

L258: Strange sentence construction.

The sentence has been removed. 

L264-265: “a single wildlife detection dog only”, I would remove “only”.

The sentence has been removed during the re-wording of the Discussion.

L271: I would change “mortality” with “enhanched/augmetned/increase in mortality” since mortality in general is a natural phenomenon.

The sentence has been amended during the re-wording of the Discussion.

L271-273: “and due to limited battery duration at required small transmitter sizes is presently unable to locate newts over periods longer than a few weeks after breeding” I think something is missing in this sentence.

The sentence has been amended. 

L275-276: Something is wrong with the punctation, or some word is missing.

The sentence has been amended. 

L276: I would remove “for example”, same at line 284, 287, 317.

The sentence has been amended. 

L279: And also useful for monitoring purposes I suppose.

The sentence has been amended accordingly. 

L286: “individuals of”

The section has been removed during the re-wording of the Discussion. 

L288-290: Please rephrase and try to merge the two exapmples.

The text has been removed during the re-wording of the Discussion.

L294: Maybe better “wide availability”.

The text has been removed during the re-wording of the Discussion.

L299-301: Strange construction, please rephrase.

The text has been removed during the re-wording of the Discussion. 

L324: Here and above, why “Sandy Soil Full” is capitalized?

We capitalised this phrase (and other similar phrases elsewhere) because it refers to the name of a treatment. We are however more than happy to remove the capitals if deemed more appropriate. 

L326: And another main point could be to use more dogs to assess the variability of detection dogs’ ability to locate newts.

We are grateful for the constructive and useful comment, but would prefer to address the topic in a future manuscript which will specifically focus on a comparison between different dogs. 

L478: “B” should not be capitalized.

This has been amended.

---

## [Decision Letter · Decision Letter 1]

16 Apr 2023

An experimental assessment of detection dog ability to locate great crested newts (Triturus cristatus) at distance and through soil

PONE-D-22-22016R1

Dear Dr. Glover,

We’re pleased to inform you that your manuscript has been judged scientifically suitable for publication and will be formally accepted for publication once it meets all outstanding technical requirements.

Kind regards,

Christopher Walton, Ph.D.

Academic Editor

PLOS ONE

Additional Editor Comments (optional):

Reviewers' comments:

Reviewer's Responses to Questions

**Comments to the Author**

1. If the authors have adequately addressed your comments raised in a previous round of review and you feel that this manuscript is now acceptable for publication, you may indicate that here to bypass the “Comments to the Author” section, enter your conflict of interest statement in the “Confidential to Editor” section, and submit your "Accept" recommendation.

Reviewer #2: All comments have been addressed

2. Is the manuscript technically sound, and do the data support the conclusions?

Reviewer #2: Yes

3. Has the statistical analysis been performed appropriately and rigorously? 

Reviewer #2: Yes

4. Have the authors made all data underlying the findings in their manuscript fully available?

Reviewer #2: Yes

5. Is the manuscript presented in an intelligible fashion and written in standard English?

Reviewer #2: Yes

6. Review Comments to the Author

Reviewer #2: General comments:

I am reviewing this manuscript for the second time, and I acknowledge the work of the authors in improving the manuscript following reviewers’ comments.

One last concern is: the authors say that data are available on Dryad, however, I was not able to find any doi/link.

At this point, I only have a few minor concerns:

Line 41: I think that “migrate” is more appropriate than “disperse” here, see 1. Cayuela H, Valenzuela-Sánchez A, Teulier L, Martínez-Solano Í, Léna J-P, Merilä J, et al. Determinants and Consequences of Dispersal in Vertebrates with Complex Life Cycles: A Review of Pond-Breeding Amphibians. Quartely Rev Biol. 2020;95: 1–36. doi:10.1086/707862

L45: And also the introduction of fish (e.g., Denoël M, Perez A, Cornet Y, Ficetola GF. Similar local and landscape processes affect both a common and a rare newt species. PLoS One. 2013;8: e62727. doi:10.1371/journal.pone.0062727; Falaschi M, Muraro M, Gibertini C, Delle Monache D, Lo Parrino E, Faraci F, et al. Explaining declines of newt abundance in northern Italy. Freshw Biol. 2022;67: 1174–1187. doi:10.1111/FWB.13909;).

L140: “the dog indicated, confirmed with the testers who were out of view.” not clear.

L149: “hygrometer thermometer” thermo-hygrometer?

L243-246: The underlying maths is not very clear here.

L373-377: I think this type of “control” is a very hard challenge for the dog/handler team. This implies that the authors tested dog and handler abilities in the worst-case scenario, possibly inflating false positives. I would stress more, if the authors think it is adequate, that this is a strength more than a weakness of this test.

7. PLOS authors have the option to publish the peer review history of their article (what does this mean?). If published, this will include your full peer review and any attached files.

Reviewer #2: No

---

## [Editor Report · Acceptance letter]

27 Apr 2023

PONE-D-22-22016R1 

An experimental assessment of detection dog ability to locate great crested newts (*Triturus cristatus*) at distance and through soil 

Dear Dr. Glover:

I'm pleased to inform you that your manuscript has been deemed suitable for publication in PLOS ONE. Congratulations! Your manuscript is now with our production department. 

Kind regards, 

on behalf of

Dr. Christopher Walton 

Academic Editor

PLOS ONE